# Antibacterial Activity of Chromomycins from a Marine-Derived *Streptomyces microflavus*

**DOI:** 10.3390/md18100522

**Published:** 2020-10-21

**Authors:** Eunji Cho, Oh-Seok Kwon, Beomkoo Chung, Jayho Lee, Jeongyoon Sun, Jongheon Shin, Ki-Bong Oh

**Affiliations:** 1Department of Agricultural Biotechnology, College of Agriculture and Life Sciences, Seoul National University, Seoul 08826, Korea; eunji525@snu.ac.kr (E.C.); beomkoo01@snu.ac.kr (B.C.); jayho@snu.ac.kr (J.L.); rhdlfdhkdsns@snu.ac.kr (J.S.); 2Natural Products Research Institute, College of Pharmacy, Seoul National University, Seoul 08826, Korea; ideally225@snu.ac.kr

**Keywords:** chromomycins, marine actinomycete, *Streptomyces microflavus*, antimicrobial activity, MRSA, resistance

## Abstract

A marine-derived actinomycete (*Streptomyces* sp. MBTI36) exhibiting antibacterial activities was investigated in the present study. The strain was identified using genetic techniques. The 16S rDNA sequence of the isolate indicated that it was most closely related to *Streptomyces microflavus*. Furthermore, a new chromomycin A_9_ (**1**), along with chromomycin Ap (**2**), chromomycin A_2_ (**3**), and chromomycin A_3_ (**4**), were isolated from the ethyl acetate extract. Their structures were determined using extensive spectroscopic methods including 1D and 2D NMR, and HRMS, as well as comparisons with previously reported data. Compounds **1**–**4** showed potent antibacterial activities against Gram-positive bacteria including methicillin-resistant *Staphylococcus aureus* (MRSA). During a passage experiment, minimum inhibitory concentration (MIC) values for compounds **1**–**4** showed no more than a 4-fold increase from the starting MIC value, indicating that no resistance was detected over the 21 passages.

## 1. Introduction

Antibiotic resistance is an increasing threat to global health [1]. The development of new antimicrobial drugs is a priority to combat the increasing spread of antibiotic-resistant bacteria [2,3]. Since the discovery of antibiotics, the majority of new antibiotics have been isolated from actinomycetes, especially *Streptomyces* spp.; however, the number of newly discovered antibiotics continues to decrease [4]. To solve this problem, many scientists have focused on marine microorganisms. The deep sea environment provides unique growth conditions for microorganisms, such as minimal light, high pressure, salinity, extreme temperature, and lack of oxygen [5,6]. The different environment might confer unusual features on marine actinomycetes; thus, in comparison with their terrestrial counterparts, new bioactive secondary metabolites could be produced [7]. Therefore, actinomycetes from marine habitats can be a rich source of structurally unique natural compounds with diverse biological activities, including antibacterial and anticancer activities [8].

Chromomycins are members of the aureolic acid family, which were first isolated from *Streptomyces griseus* No. 7 (ATCC No. 13273) [9]. Members of this family, which include mithramycin, olivomycin, durhamycin, and chromomycin, are glycosylated aromatic polyketides with potent antitumor and antibacterial activities [10]. These polyketides have a tricyclic aglycone core with two aliphatic side chains at C-3 and C-7 and two sugar chains at C-2 and C-6. The compounds are distinguished by the type of sugar; chromomycins have a disaccharide chain consisting of two d-olioses at C-2 and a trisaccharide chain consisting of two D-olivoses and L-chromose. Chromomycins possess a broad range of biological activities such as antimicrobial [10,11], anticancer [12,13,14], and antiviral [15] activities, and are found in terrestrial and marine bacteria of the genus *Streptomyces*. Similar to other aureolic acid family members, chromomycins interact with the DNA helix minor groove in regions with high GC content and in a non-intercalative way with Mg^2+^ cations, causing DNA damage in treated cells [16,17,18]. 

When searching for bioactive secondary metabolites from marine-derived actinomycetes, we characterized a strain, MBTI36, collected from marine sediment from Jeju Island, Korea, and identified it as *Streptomyces microflavus* based on 16S rDNA analysis. An organic extract of a liquid culture of this strain exhibited potent antibacterial activity, with a minimum inhibitory concentration (MIC) value of 0.5 μg/mL, against *Staphylococcus aureus*, a Gram-positive pathogenic bacterium. Antibacterial activity-guided separation of the extract using diverse chromatographic methods led to the isolation of a new chromomycin derivative A_9_ (**1**), with the known chromomycins Ap (**2**), A_2_ (**3**), and A_3_ (**4**). Herein, the structure of a new chromomycin A_9_ (**1**) and the antibacterial activity spectrum of compounds **1**–**4** against several representative pathogenic microorganisms including methicillin-resistant *Staphylococcus aureus* (MRSA) are reported.

## 2. Results

### 2.1. Taxonomy and Phylogenetic Analysis of MBTI36

The 16S rDNA of strain MBTI36 was amplified using polymerase chain reaction (PCR) and sequenced (1389 bp). After a basic logic alignment search tool (BLAST) sequence comparison, strain MBTI36 showed 99.9% identity to *Streptomyces microflavus* NBRC13062 (type strain, GenBank accession number: NR1123524), *S. microflavus* strain DR009 (GenBank accession number: JQ422181), strain PM122 (GenBank accession number: JQ422127), and strain PM100 (GenBank accession number: JQ422182). Thus, this strain was designated as *Streptomyces microflavus* strain MBTI36 (GenBank accession number: MK396664). The phylogenetic tree generated using the neighbor-joining and maximum likelihood method based on the 16S rDNA sequence revealed the evolutionary relationships of MBTI36 with a group of known *Streptomyces* type strains (Figure 1).

### 2.2. Isolation and Structural Elucidation of Compounds ***1**–**4***

Strain MBTI36 was cultured in GTYB (10 g of glucose, 2 g of tryptone, 1 g of yeast extract, and 1 g of beef extract in 1 L of artificial seawater) broth and cultivated for 14 days without shaking. The culture filtrate was lyophilized and extracted with methanol. After evaporation of the solvent, the crude extract was separated using solvent-partitioning followed by semi-preparative high-performance liquid chromatography (HPLC) to yield four compounds. Based on combined spectroscopic analyses, including ^1^H, ^13^C nuclear magnetic resonance (NMR), two-dimensional (2D) NMR spectral analyses (COSY, HMQC, and HMBC), and UV data, compounds **2**–**4** were identified as chromomycin Ap (**2**) [19], chromomycin A_2_ (**3**) [20,21], and chromomycin A_3_ (**4**) [9,22] (Figure 2). The spectroscopic data for these compounds were in good agreement with previous reports.

The molecular formula of compound **1** was determined to be C_58_H_84_O_26_ based on high-resolution fast atom bombardment mass spectrometry (HR-FAB-MS) analysis ([M + Na]^+^
*m/z* 1219.5143, calcd for C_58_H_84_O_26_Na, 1219.5149) having 17 degrees of unsaturation. The ^13^C NMR data of this compound showed signals of two ketone carbons at δ_C_ 211.3 and 202.3, which were supported by a strong absorption band at 1722 cm^−1^ in the IR data. Similarly, two additional carbonyl carbons at δ_C_ 174.6 and 171.7, in conjunction with the absorption band at 1731 cm^−1^ in the IR data, were assigned as ester carbons. The ^13^C NMR data showed 10 deshielded carbons in the region of δ_C_ 159.9–101.0 with the corresponding proton signals at δ_H_ 6.76 and 6.64 in the ^1^H NMR data (Table 1), indicative of a naphthalene-type aromatic moiety. In addition, more than 20 oxygen-bearing methines and methylenes were observed, indicating the presence of several sugar units. For these, the five anomeric carbons at δ_C_ 100.5–95.4 and the corresponding protons at δ_H_ 5.21–4.60 in the ^13^C and ^1^H NMR data, respectively, defined the presence of five sugars. Because these preliminary interpretations accounted for 16 degrees of unsaturation (4 for carbonyls, 7 for naphthalene, and 5 for sugars), compound **1** must possess an additional cycle for the 17 degrees of unsaturation inherent from the mass data. All the structural features coincide well with congener chromomycins Ap (**2**), A_2_ (**3**), and A_3_ (**4**), indicating that **1** is a new member of this class. 

Based on this information, the structure of **1** was determined by comparing the ^13^C and ^1^H NMR data with other chromomycins as well as combined 2D NMR analyses (Figure 3, Appendix A). First, the aromatic carbons and protons had almost the same chemical shifts as congeners assigned the *tri*-oxygenated methylnaphthalene moiety (C-4-C-10, C-4a, C-8a, C-9a, and C-10a) in **1**. This interpretation was confirmed based on ^1^H-^1^H COSY, HSQC, and HMBC data including the long-range correlations of key protons (H-5, H-10, and 7-H_3_) with neighboring carbons. The expansion of this moiety was also determined based on combined COSY (H-2-H-3-H_2_-4) and HMBC correlations (H-2/C-1, H_2_-4/C-4a, and H-5/C-4) to construct a dihydroxy-3,4-dihydroanthracen-1-one moiety, the common aglycone of the chromomycin class. Then, an additional COSY correlation of H-3 placed a methoxy-bearing methine (C-1′) at this position. A COSY-based 2,3-dihydroxypropyl group (C-3′-C-5′) was connected at this methine via a ketone (C-2′) based on HMBC correlations (H-1′/C-2′ and H-3′/C-2′), establishing a five carbon side chain that was previously reported for the chromomycin family.

The ^13^C and ^1^H NMR and 2D NMR data for the sugar portion of **1** were also similar to **4**, indicating the same sugar identities and connectivity (Table 1). Consequently, the five sugar units were identified as two D-olioses, two D-olivoses, and one L-chromose, based on the comparison of NMR data with **4** using COSY correlations among the sugar protons. Subsequently, the linkage between the sugar moieties and their connection at the aglycone were secured by 3-bond HMBC correlations of anomeric carbons with protons (Figure 3). However, an ethyl group was identified based on the COSY and HSQC data (δ_C_ 27.8, δ_H_ 2.46; δ_C_ 9.7, δ_H_ 1.19). The attachment of this group at C-A4 of D-oliose (sugar A) via a ketone (δ_C_ 174.6) was also found based on a series of HMBC correlations, replacing the 4-*O*-acetyl substituent of **4** with the 4-*O*-propioyl group in **1**. Thus, the structure of compound **1** (chromomycin A_9_) was determined to be a new chromomycin possessing a 4-*O*-propioyl-D-oliose moiety.

### 2.3. Antimicrobial Activity of Compounds ***1**–**4***

The antimicrobial activities of compounds **1**–**4** were evaluated against phylogenetically diverse pathogenic bacterial and fungal strains. These compounds displayed significant antibacterial activity against the Gram-positive strains tested (*Staphylococcus aureus* ATCC25923, *Enterococcus faecium* ATCC19434, and *E*. *faecalis* ATCC19433), with MIC values of 0.03–0.5 μg/mL (Table 2). However, the compounds did not show inhibitory activity against Gram-negative bacteria such as *Klebsiella pneumoniae* ATCC10031 and *Escherichia coli* ATCC25922 (MIC > 128 μg/mL), except for *Salmonella enterica* ATCC14028 (MIC = 0.5–1 μg/mL) when using ampicillin and tetracycline as positive control compounds. The antifungal activities of compounds **1**–**4** were also evaluated against pathogenic fungal strains, including *Candida albicans* ATCC10231, *Aspergillus fumigatus* HIC6094, *Trichophyton rubrum* NBRC9185, and *Trichophyton mentagrophytes* IFM40996, using amphotericin B as a positive control compound. However, these compounds did not exhibit inhibitory activity against the tested fungi. Based on the data, compounds **1**–**4** were further evaluated against drug-resistant *S*. *aureus* strains. These compounds exhibited potent antibacterial activities against all methicillin-sensitive *S*. *aureus* (MSSA) and MRSA strains (Table 3). Notably, compounds **1**–**4** showed significant broad-spectrum antibiotic effects on MRSA strains with MIC values of 0.06–0.25 μg/mL, which are more potent than the values of major classes of antibiotics that include daptomycin (MIC > 32 μg/mL), vancomycin (MIC = 0.5–2 μg/mL), platensimycin (MIC = 4–8 μg/mL), linezolid (MIC = 1–2 μg/mL), and ciprofloxacin (MIC = 0.13– > 32 μg/mL).

### 2.4. Multi-Step Resistance Development

The potential for resistance development to compounds **1**–**4** in a MRSA strain was also investigated. Representative resistance development profiles for ciprofloxacin (a comparator antibiotic) and compounds **1**–**4** against *S*. *aureus* ATCC43300 are presented in Figure 4. In the present study, resistance was defined as a >4-fold increase in the initial MIC [23,24]. A steady increase in MIC was observed for ciprofloxacin during the passage experiment in *S*. *aureus* ATCC43300, with a final MIC of 32 μg/mL, a 128-fold change compared with the initial MIC value (MIC = 0.25 μg/mL). Conversely, for compounds **1**–**4**, a 2-fold increase in the MIC (from 0.13 to 0.25 μg/mL, from 0.13 to 0.25 μg/mL, from 0.06 to 0.13 μg/mL, and from 0.13 to 0.25 μg/mL, respectively) was observed for *S*. *aureus* ATCC43300 after 21 passages. During the passage experiment, MIC values for compounds **1**–**4** showed no more than a 4-fold increase from the starting MIC value, indicating that resistance did not develop during the 21 passages.

## 3. Discussion

Chromomycins, members of the aureolic acid family, are tricyclic glycosylated polyketides with a broad range of bioactivity spectrums including antimicrobial and anticancer activity that are produced by terrestrial and marine *Streptomycete* species. In the present study, a chemical investigation of a liquid culture extract of the marine-derived actinomycete *S*. *microflavus* strain MBTI36 led to the isolation of four chromomycin derivatives (**1**–**4**). The structures of compounds **2**–**4** were elucidated based on spectroscopic data and comparisons with previously reported data [9,19,20,21,22], and were identified as chromomycin Ap (**2**), chromomycin A_2_ (**3**), and chromomycin A_3_ (**4**). The ^13^C and ^1^H NMR spectroscopic data of compound **1** were similar to congener compounds **2**–**4**, indicating **1** as a new member of this class. Similar to compound **4,** the linkage among the sugar moieties and their attachments to the aglycone were the same; however, the COSY and HSQC data showed an ethyl group in compound **1**. The attachment of this group at C-A4 of D-oliose (sugar A) via a ketone was also found based on a series of HMBC correlations, replacing the 4-*O*-acetyl substituent of **4** with the 4-*O*-propioyl group in **1**. Thus, the structure of compound **1** was determined to be a new chromomycin A_9_ possessing a 4-*O*-propioyl-D-oliose moiety. 

Chromomycins, especially chromomycin A_3_ (**4**), were previously isolated from *Streptomycete* species including *S*. *griseus* and *S*. *cavourensis* [9,10] and showed antibacterial activity against Gram-positive bacteria such as *Bacillus subtilis*, *S*. *aureus*, and *Enterococcus hirae* [11]. In the present study, the isolated compounds **1**–**4** displayed significant antibacterial activity against Gram-positive pathogenic strains such as *S*. *aureus*, *E*. *faecium*, and *E*. *faecalis*, with MIC values of 0.03–0.5 μg/mL. These compounds also exhibited potent inhibitory activities against MRSA strains, with MIC values of 0.06–0.25 μg/mL, which are more potent than values of major classes of antibiotics that include vancomycin and linezolid. The potential for development of resistance to compounds **1**–**4** in the MRSA strain *S*. *aureus* ATCC43300 was also investigated. As shown in Table 3, ciprofloxacin showed potent antibacterial activities against MRSA strains *S*. *aureus* ATCC43300 and ATCC700787. However, based on multi-step (21-passage) resistance selection studies in the presence of sub-inhibitory concentrations of ciprofloxacin (a comparator antibiotic), a steady increase in MIC for ciprofloxacin against *S*. *aureus* ATCC43300 during the passage experiment was observed (Figure 4). Reportedly, topoisomerase IV is the primary target of fluoroquinolones in *S*. *aureus*, which may be due to the greater sensitivity of topoisomerase IV, relative to DNA gyrase, to these agents [25]. Furthermore, resistance from an altered DNA gyrase requires resistant topoisomerase IV for expression [25,26]. By contrast, during the passage experiment, MIC values for compounds **1**–**4** showed no more than a 4-fold increase from the starting MIC value, indicating resistance did not develop during the 21 passages. 

Members of the aureolic acid family of antibiotics including chromomycins were isolated in the present study based on their antimicrobial activity against Gram-positive bacteria. However, the compounds were not active against Gram-negative bacteria due to permeability problems; thus, the main pharmacological interest resides in their antitumor activity [10]. In several studies, the antitumor activities and mode of action of aureolic acid family antibiotics was shown to be partly due to a non-intercalating interaction with GC regions in the minor groove of the DNA double helix by forming dimeric complexes with Mg^2+^ [16,17,18]. A suggested mechanism of action involves the binding of chromomycin A_3_ to DNA as a Mg^2+^ dimer, which cross-links the two strands, thus blocking the processes of replication and mainly, transcription of DNA, resulting in cytotoxic activity. Therefore, modifications of the side chain can potentially be exploited for modulation of the biological activity of chromomycin compounds. Chromomycins derived from marine actinomycetes should be further investigated.

## 4. Materials and Methods 

### 4.1. General Experimental Equipments

Optical rotations were measured using a JASCO P-1020 polarimeter (Jasco, Tokyo, Japan) with a 1 cm cell. UV spectra were acquired using a Hitachi U-3010 spectrophotometer (Hitachi, Tokyo, Japan). IR spectra were recorded on a JASCO 4200 FT-IR spectrometer (JASCO, Easton, MD, USA) using a ZnSe cell. NMR spectra were recorded in CDCl_3_ with the solvent peaks (δ_H_ 7.26/δ_c_ 77.2) as internal standards on a JEOL JNM 400 MHz NMR spectrometer (JOEL, Peabody, MA, USA). HR-FAB-MS data were obtained at the National Center for Inter-University Research Facilities (NCIRF), Seoul National University, and acquired using a JEOL JMS 700 mass spectrometer (Jeol, Tokyo, Japan) with 6 keV-energy, emission current 5.0 mA, xenon as inert gas, and meta-nitrobenzyl alcohol as the matrix. HPLC separations were performed on a SpectraSYSTEM p2000 equipped with a refractive index detector (SpectraSYSTEM RI-150; Thermo Scientific, Waltham, MA, USA) and a UV-Vis detector (Gilson UV-Vis-151, Gilson, Middleton, WI, USA). All solvents used were of spectroscopic grade or were distilled prior to use.

### 4.2. Taxonomic Identification of the Chromomycin Producing Microorganism

Marine sediment samples were collected from the shoreline of Jeju Island, Korea. The air-dried sediment (1 g) was mixed with 10 mL of sterilized artificial seawater and shaken at 150 rpm for 30 min at 25 °C. The suspension was serially diluted with sterilized artificial seawater and 0.1 mL volumes of three appropriate dilutions, based on turbidity, were spread onto actinomycete isolation agar plates supplemented with cycloheximide (100 μg/mL) and artificial seawater. Plates were incubated at 28 °C for 2 to 3 weeks. To obtain single strains, colonies were transferred several times on fresh agar plates by streaking.

The isolated bacterial strain MBTI36 was identified using standard molecular biological protocols including DNA amplification and sequencing of the 16S rDNA region. Briefly, genomic DNA was prepared from MBTI36 mycelium using an i-Genomic BYF DNA Extraction Mini Kit (Intron Biotechnology, Seoul, Korea) according to the manufacturer’s protocol. PCR amplification using the primers 27F (5′-AGAGTTTGATCCTGGCTCAG-3′) and 1429R (5′- GGTTACCTTGTTACGACTT-3′) was performed under the following conditions: pre-denaturation (95 °C, 5 min), 30 cycles of denaturation (95 °C, 15 s), annealing (50 °C, 30 s), extension (72 °C, 1 min 30 s), and final extension (72 °C, 5 min). The nucleotide sequence was deposited in GenBank under accession number MK396664 and aligned according to nucleotide BLAST (Basic Local Alignment Search Tool, version 2.10) of the National Center for Biotechnology Information (NCBI, Bethesda, MD, USA) database. The phylogenetic tree was simulated using a neighbor-joining method [27] and evolutionary distances were analysed with the Kimura two-parameter model using MEGA-X software (Molecular Evolutionary Genetics Analysis, version 10, https://megasoftware.net) [28]. The bootstrap was replicated a thousand times.

### 4.3. Cultivation 

MBTI36 was sporulated on GTYB agar plates at 28 °C for 5 days. Mature spores were inoculated in 25 mL GTYB broth at 28 °C for 24 h on a rotatory shaker. Each seed culture was transferred directly to 100 mL of fresh GTYB broth (each flask containing 125 mL of culture broth, 160 flasks used) and incubated at 28 °C for 14 days without shaking.

### 4.4. Extraction and Isolation

Culture broths were collected from 160 flasks (total 20 L) and filtered through filter paper. Culture filtrate was then lyophilized and extracted three times with 5 L of methanol. After removing the solvent under vacuum, the crude extract was redissolved in water and partitioned with *n*-hexane (0.4 g) and ethyl acetate (1.83 g). Based on the results of the antibacterial activity assay, the latter fraction was separated using semi-preparative reverse-phase HPLC (Agilent C_18_ column, 10 × 250 mm; 2.0 mL/min; water-acetonitrile, 50:50) eluting compounds **4** (*t*_R_ = 15.8 min), **2** (*t*_R_ = 22.6 min), **1** (*t*_R_ = 23.4 min), and **3** (*t*_R_ = 34.5 min) in that order. Further purification for compounds **1** and **2** was accomplished using analytical HPLC (Agilent C_18_ column, 4.6 × 250 mm; 0.7 mL/min; water-acetonitrile, 55:45). The metabolites were isolated in the following amounts: 6.0, 8.7, 14.1, and 130.0 mg of **1**–**4**, respectively.

Chromomycin A_9_ (**1**): pale yellow, amorphous solid; [α]${}_{D}^{25}$ –18.3 (*c* 0.25, MeOH); UV (MeOH) λ_max_ (log ε) 205 (3.78), 223 (2.14), 282 (1.93), 317 (1.22), 329 (1.19), 430 (1.34) nm; IR (ZnSe) ν_max_ 3408, 2933, 1731, 1722, 1252, 1167, 1068 cm^−1^; ^13^C and ^1^H NMR data, see Table 1; HR-FAB-MS *m/z* 1219.5143 [M + Na]^+^ (calcd for C_58_H_84_O_26_Na, 1219.5149).

### 4.5. Antibacterial Activity Assays

The antibacterial activity assays were performed according to the Clinical and Laboratory Standards Institute (CLSI) guide methods [29]. Gram-positive bacteria (*S*. *aureus* ATCC25923, *E*. *faecium* ATCC19434, and *E*. *faecalis* ATCC19433) and Gram-negative bacteria (*S*. *enterica* ATCC14028, *K*. *pneumoniae* ATCC10031, and *E*. *coli* ATCC25922) were used for each quality control strain. The following drug-resistant strains were obtained from the stock Culture Collection of Antimicrobial Resistant Microorganisms (CCARM; Seoul Women’s University) and American Type Culture Collection (ATCC) and used for antibacterial activity assays: MSSA strains were CCARM0027, CCARM0204, CCARM0205, and CCARM3640; MRSA strains were CCARM3089, CCARM3090, CCARM3634, CCARM3635, ATCC43300, ATCC700787, and ATCC700788. Cells were cultured overnight in Mueller–Hinton broth (MHB; BD Difco, Sparks, MD, USA) at 37 °C, collected by centrifugation, and washed twice with sterile distilled water. Each test compound was dissolved in dimethyl sulfoxide (DMSO) and diluted with MHB to prepare serial 2-fold dilutions ranging from 0.008 to 128 μg/mL. The final DMSO concentration was maintained at 1% by adding DMSO to the medium according to CLSI guidelines. In each well of a 96-well plate, 90 μL of MHB containing the test compound was mixed with 10 μL of broth containing approximately 5 × 10^6^ colony-forming units (cfu)/mL of test bacterium (final concentration: 5 × 10^5^ cfu/mL) adjusted to match the turbidity of a 0.5 MacFarland standard at 625 nm wavelength. The plates were incubated for 24 h at 37 °C. The MIC was defined as the lowest concentration of test compound that prevented cell growth. Ampicillin (Duchefa, Haarlem, The Netherlands), tetracycline, daptomycin, vancomycin, platensimycin, linezolid, and ciprofloxacin (Sigma-Aldrich, St. Louis, MO, USA) were used as reference compounds.

### 4.6. Antifungal Activity Assays

The antifungal activity assays were performed in accordance with the guidelines in CLSI document M38 [30]. *C*. *albicans* ATCC10231 was cultured on potato dextrose agar (PDA) plates. After incubation for 24 h at 28 °C, yeast cells were harvested by centrifugation and washed twice with sterile distilled water. Filamentous fungi (*A*. *fumigatus* HIC6094, *T*. *rubrum* NBRC9185, and *T*. *mentagrophytes* IFM40996) were cultured on PDA plates at 28 °C for 5 days. Spores were harvested and washed twice with sterile distilled water. Stock solutions of the compound were prepared in DMSO. Each stock solution was diluted in Roswell Park Memorial Institute (RPMI) 1640 broth (Sigma-Aldrich) at a concentration ranging from 0.008 to 128 μg/mL. The final DMSO concentration was maintained at 1% by adding DMSO to the broth. In each well of a 96-well plate, 90 μL of RPMI 1640 containing the test compound was mixed with 10 μL of broth containing approximately 5 × 10^5^ spores/mL of test fungus (final concentration: 5 × 10^4^ spores /mL) adjusted to match the turbidity of a 0.5 McFarland standard. The plates were incubated for 24 h (for *C*. *albicans*), 48 h (for *A*. *fumigatus*), or 96 h (for *T*. *rubrum* and *T*. *mentagrophytes*) at 35 °C. The MIC value was determined as the lowest concentration of test compound that fully inhibited cell growth. A culture with DMSO (1%) was used as a solvent control, and a culture supplemented with amphotericin B (Sigma-Aldrich) was used as a positive control.

### 4.7. Multi-Step Resistance Development Assays

Multi-step resistance development experiments were performed based on methods previously described [31] with minor modifications. The initial inoculum (10^6^ cfu/mL) of *S*. *aureus* ATCC43300, a MRSA strain, was prepared in MHB at 35 °C, and adjusted to 0.5 MacFarland standard. A dilution series of 12 concentrations was prepared for ciprofloxacin and compounds **1**–**4** in fresh MHB based on the previous MIC at 2 × the required final concentration. Next, 50 μL of bacterial inoculum was added to 50 μL of each compound dilution series in a 96-well plate (final concentration: 5 × 10^5^ cfu/mL) and a control plate (no antimicrobial). The plates were incubated at 35 °C for 24 h and the MIC was determined. The cultures were passaged daily for 21 days using 0.5-mL inocula (5 × 10^5^ cfu/mL) from the 96-well plate corresponding to 0.5× the MIC determined from the previous passage to inoculate a fresh series of compound dilutions and control plates. 

## Figures and Tables

**Figure 1 marinedrugs-18-00522-f001:**
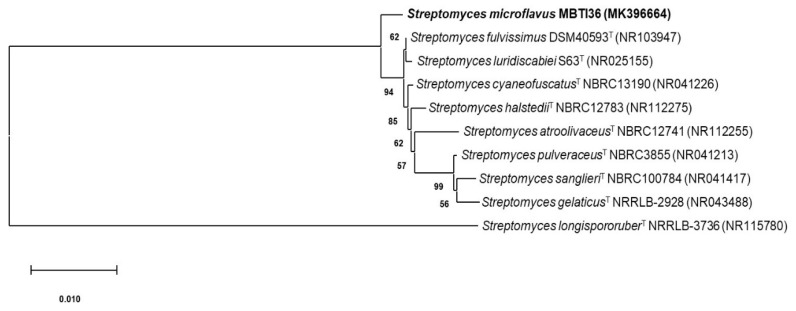
A neighbor-joining phylogenetic tree of strain MBTI36 based on 16S rDNA sequence. The phylogenetic tree was constructed using MEGA-X software (Molecular Evolutionary Genetics Analysis, version 10) and bootstrap was replicated a thousand times. The Kimura two-parameter model considering transversional and transitional substitution rates was used to measuring distance. Bar indicates 10 nucleotide substitutions per 1000 sites. T: type strain.

**Figure 2 marinedrugs-18-00522-f002:**
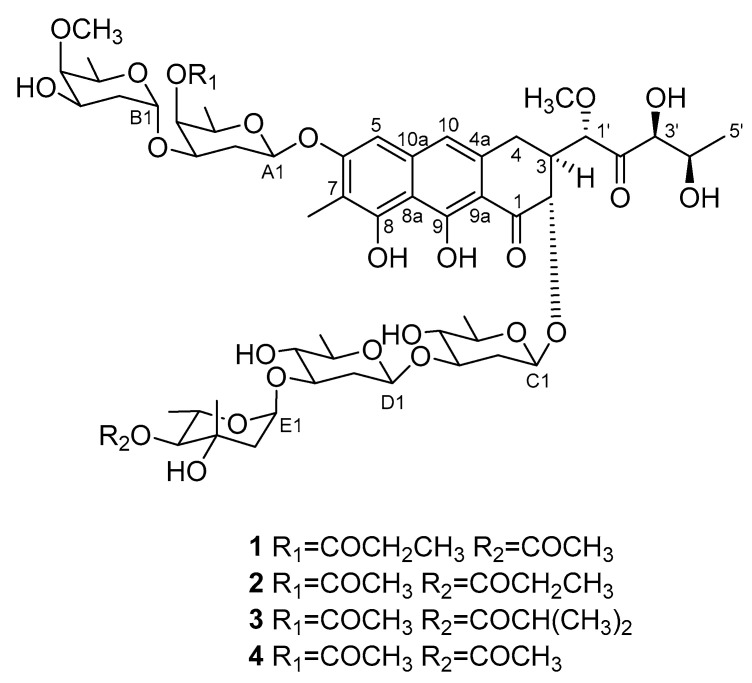
The structures of compounds **1**–**4**.

**Figure 3 marinedrugs-18-00522-f003:**
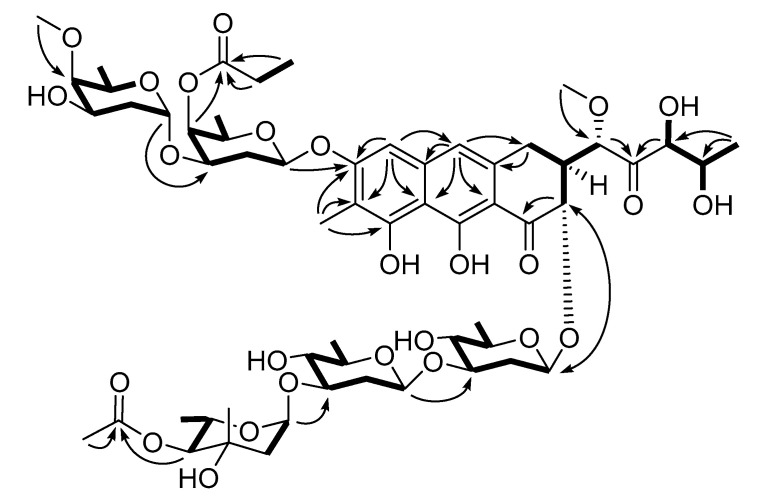
Key correlations of COSY (bold) and HMBC (arrows) experiments for compound **1**.

**Figure 4 marinedrugs-18-00522-f004:**
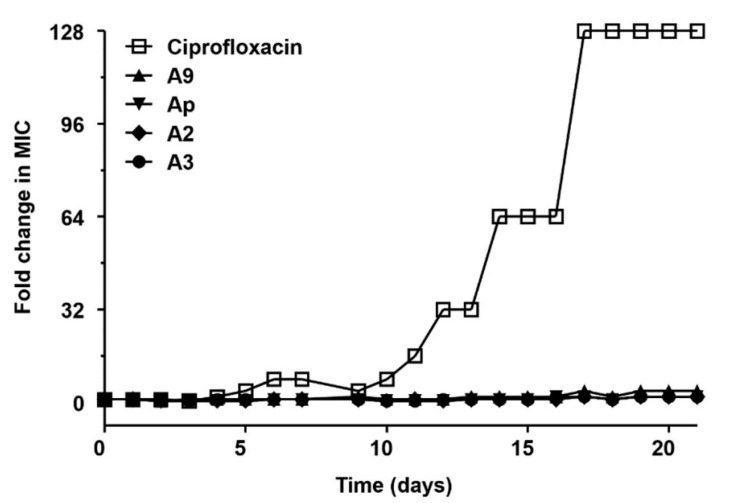
Resistance acquisition during 21 serial passaging (21 days) in the presence of sub-MIC (0.5 × the MIC determined from the previous passage) levels of ciprofloxacin, chromomycin A_9_ (**1**), Ap (**2**), A_2_ (**3**), and A_3_ (**4**) for *S. aureus* ATCC43300. The *y* axis is the highest concentration the cells grew to during passaging. For ciprofloxacin, 128 × MIC was the highest concentration tested. The figures are representative of three independent experiments.

**Table 1 marinedrugs-18-00522-t001:** ^13^C and ^1^H NMR assignments for compound **1** in CDCl_3_
^a^.

Position	δ_C_, Type	δ_H_ (*J* in Hz)	Position	δ_C_, Type	δ_H_ (*J* in Hz)
1	202.3, C		4-*O*-methyl-d-oliose (Sugar B)		
2	76.1, CH	4.72, d (11.4)	B1	95.4, CH	5.11, br s
3	44.0, CH	2.60, m	B2	33.7, CH_2_	1.76, m; 1.73, m
4	27.1, CH_2_	3.10, m	B3	66.0, CH	3.96, m
		2.67, dd (16.6, 3.5)	B4	81.7, CH	3.22, d (2.8)
5	101.0, CH	6.64, s	B5	66.9, CH	3.87, q (6.6)
6	159.9, C		B6	17.4, CH	1.28, d (6.3)
7	111.9, C		B4-OCH_3_	62.6, CH_3_	3.60, s
8	156.3, C		d-olivose (Sugar C)		
9	165.5, C		C1	100.5, CH	5.10, dd (9.7, 1.3)
10	117.2, CH	6.75, s	C2	37.7, CH_2_	2.48, m; 1.70, m
4a	134.8, C		C3	82.5, CH	3.61, m
8a	108.3, C		C4	75.3, CH	3.12, m
9a	108.3, C		C5	72.3, CH	3.38, m
10a	138.6, C		C6	18.2, CH_3_	1.35, d (5.6)
7-CH_3_	8.4, CH_3_	8.40, s	d-olivose (Sugar D)		
8-OH		9.81, s	D1	99.9, CH	4.60, dd (9.6, 1.7)
1′	82.0, CH	4.70, d (1.5)	D2	37.3, CH_2_	2.28, m; 1.70, m
2′	211.3, C		D3	80.9, CH	3.50, m
3′	78.3, C	4.22, br s	D4	75.4, CH	3.12, m
4′	68.1, C	4.36, m	D5	72.5, CH	3.30, m
5′	20.8, CH_3_	1.37, d (5.6)	D6	18.0, CH_3_	1.24, d (6.1)
1′-OCH_3_	59.9, CH_3_	3.52, s	l-chromose (Sugar E)		
4-*O*-propioyl-d-oliose (Sugar A)			E1	97.3, CH	5.02, dd (3.6, 1.8)
A1	97.6, C	5.21, dd (9.7, 2.0)	E2	43.9, CH_2_	2.04, m; 2.00, m
A2	33.2, CH_2_	2.19, m; 2.05, m	E3	70.8, C	
A3	70.2, CH	3.98, m	E4	79.9, CH	4.61, d (9.3)
A4	67.2, CH	5.18, d (2.9)	E5	67.2, CH	3.98, m
A5	70.0	3.82, q (6.5)	E6	18.0, CH_3_	1.38, d (5.4)
A6	17.0, C	1.28, d (6.3)	E3-CH_3_	23.2, CH_3_	1.35, s
*C*OCH_2_CH_3_	174.6, C		*C*OCH_3_	171.7, C	
CO*CH_2_*CH_3_	27.8, CH_2_	2.46, q (7.5)	CO*CH_3_*	21.1, CH_3_	2.14, s
COCH_2_*CH_3_*	9.7, CH_3_	1.19, t (7.5)			

^a 13^C and ^1^H NMR data were obtained at 100 and 400 MHz, respectively.

**Table 2 marinedrugs-18-00522-t002:** Results of antimicrobial activity test.

Compound	MIC (μg/mL)									
	Gram (+) Bacteria			Gram (−) Bacteria			Fungi			
	A	B	C	D	E	F	G	H	I	J
**1**	0.03	0.5	0.13	0.5	>128	>128	>128	>128	>128	>128
**2**	0.13	0.5	0.13	1	>128	>128	>128	>128	>128	>128
**3**	0.06	0.5	0.06	0.5	>128	>128	>128	>128	>128	>128
**4**	0.13	0.5	0.13	0.5	>128	>128	>128	>128	>128	>128
Ampicillin	0.06	0.5	0.25	0.25	128	32				
Tetracycline	0.06	0.13	0.25	0.25	0.5	0.5				
Amphotericin B							0.5	1	1	1

A: *Staphylococcus aureus* ATCC25923, B: *Enterococcus faecium* ATCC19434, C: *Enterococcus faecalis* ATCC19433, D: *Salmonella enterica* ATCC14028, E: *Klebsiella pneumoniae* ATCC10031, F: *Escherichia coli* ATCC25922, G: *Candida albicans* ATCC10231, H: *Aspergillus fumigatus* HIC6094, I: *Trichophyton rubrum* NBRC9185, J: *Trichophyton mentagrophytes* IFM40996.

**Table 3 marinedrugs-18-00522-t003:** Antibacterial activities of compounds **1**–**4** against MSSA and MRSA strains.

Microorganism	MIC (μg/mL)								
	Dap	Van	Pla	Lin	Cip	1	2	3	4
CCARM0027 *^a^*	8	0.5	4	2	0.25	0.13	0.13	0.06	0.13
CCARM0204 *^a^*	2	0.25	4	1	0.25	0.06	0.06	0.03	0.06
CCARM0205 *^a^*	1	0.13	2	1	0.25	0.06	0.13	0.06	0.06
CCARM3640 *^a^*	8	0.25	4	2	0.25	0.13	0.25	0.06	0.13
CCARM3089 *^b^*	>32	1	8	2	>32	0.13	0.25	0.13	0.13
CCARM3090 *^b^*	>32	1	8	1	>32	0.13	0.25	0.13	0.13
CCARM3634 *^b^*	>32	0.5	8	2	>32	0.13	0.13	0.06	0.13
CCARM3635 *^b^*	>32	1	8	2	>32	0.13	0.06	0.06	0.13
ATCC43300 *^b^*	>32	1	4	2	0.25	0.13	0.13	0.06	0.13
ATCC700787 *^b^*	>32	2	8	2	0.13	0.13	0.25	0.25	0.13
ATCC700788 *^b^*	>32	2	8	2	16	0.13	0.25	0.13	0.13

*^a^* Methicillin-sensitive *Staphylococcus aureus* (MSSA). *^b^* Methicillin-resistant *Staphylococcus aureus* (MRSA). Dap: daptomycin, Van: vancomycin, Pla: platensimycin, Lin: Linezolid, Cip: ciprofloxacin.

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
