# Peer review of "Antibacterial Activity of Chromomycins from a Marine-Derived Streptomyces microflavus"

_marinedrugs, 2020, doi:10.3390/md18100522_

Round 1
Reviewer 1 Report
Authors in the manuscript “Antimicrobial Activity of Chromomycins from Marine-Derived Streptomyces microflavus” isolate and identify a marine-derived actinomycete exhibiting antibacterial activities. Moreover, they identify a new chromomycin A9, chromomycin Ap, chromomycin A2 and chromomycin A3 from the secondary metabolites. Structures were determined and antibacterial activities against Gram-positive bacteria established. Moreover, authors indicate that no resistance is developed during the 21 passages.
The work is of interest in the context of the search for new and efficient antimicrobials. However, the authors should better explain the origin of the strain and describe isolation and selection processes. The description of the strain isolation section is very imprecise. Furthermore, I personally do not find myself qualified to review the “Isolation and Structural Elucidation of Compounds 1–4” sections which constitute most of the body of the manuscript. The microbiological aspects are limited.
Authors indicated that isolate strain was identified as Streptomyces microflavus based on 16S rDNA analysis; however, in the Materials and Methods Section, they suggest that the strain was identified using standard molecular biological protocols including DNA amplification and sequencing of the internal transcribed spacer region. Could they explain it?
Why authors use ampicillin and tetracycline as positive control compounds? Explain.
Table 2. the names of the microorganisms must be written in italics.
Why ciprofloxacin is a comparator antibiotic? Explain.
Why 4.1. General Experimental Procedures section?
4.3. Fermentation? Antibiotics are secondary metabolites. I do not think this is the best reference for the process of its production.
How they get the 20 l of MBTI36 culture filtrate? Explain.
Why are these antibiotics (ampicillin, tetracycline, daptomycin, vancomycin, platensimycin, linezolid and ciprofloxacin) used as reference compounds and what are the highest concentrations used? (Sigma-Aldrich, St. Louis, MO, USA) were used.
Do they use spores or cells? Authors indicate that spores were harvested from PDA plates but test compound was mixed with 10 μL 291 of broth containing approximately 5 × 105 cfu/mL of test fungus. Explain
Author Response
Thank you very much for your careful and valuable review of our manuscript. We made all revisions and corrections as far as we could. I hope this is the right answer for your request. What follows is our response to reviewer’s critique with the explanation of the changes implemented in the paper and a rebuttal when appropriate.
Comment 1:
The work is of interest in the context of the search for new and efficient antimicrobials. However, the authors should better explain the origin of the strain and describe isolation and selection processes. The description of the strain isolation section is very imprecise.
Answer)
We appreciate Reviewer’s kind comments. According to the reviewer’s comments, we revised and provided a more detailed description of the isolation and selection processes is provided in 4.2. Taxonomic Identification of the Chromomycin Producing Microorganism section (page 8, lines 235-241): Marine sediment samples were collected from the shoreline of Jeju Island, Republic of Korea. The air-dried sediment (1 g) was mixed with 10 mL of sterilized artificial seawater and shaken at 150 rpm for 30 min at 25 °C. The suspension was serially diluted with sterilized artificial seawater and 0.1 mL volumes of 3 appropriate dilutions, based on turbidity, were spread onto actinomycete isolation agar plates supplemented with cycloheximide (100 μg/mL) and artificial seawater. Plates were incubated at 28°C for 2 to 3 weeks. To obtain single strain, colonies were transferred several times on fresh agar plates by streaking.
Comment 2:
Authors indicated that isolate strain was identified as Streptomyces microflavus based on 16S rDNA analysis; however, in the Materials and Methods Section, they suggest that the strain was identified using standard molecular biological protocols including DNA amplification and sequencing of the internal transcribed spacer region. Could they explain it?
Answer)
According to the reviewer’s comments, lines 242-247 (4.2. Taxonomic Identification of the Chromomycin Producing Microorganism section) were revised: “Strain MBTI36 was identified using standard molecular biological protocols including DNA amplification and sequencing of the 16S rDNA region. Briefly, genomic DNA was prepared from MBTI36 mycelium using an i-Genomic BYF DNA Extraction Mini Kit (Intron Biotechnology, Seoul, Republic of Korea) according to the manufacturer’s protocol. PCR amplification using the primers 27F (5′-AGAGTTTGATCCTGGCTCAG-3′) and 1429R (5′- GGTTACCTTGTTACGACTT-3′) was performed under the following conditions: ”
Comment 3:
Why authors use ampicillin and tetracycline as positive control compounds? Explain.
Answer)
In the evaluation of antibacterial activity, in general, ampicillin exhibited strong inhibitory activity against G(+) and G(-) bacteria, but weakly against Klebsiella pneumoniae and Escherichia coli as shown in Table 2. Specifically, the MIC value of ampicillin against K. pneumoniae was 128 μg/mL or >128 μg/mL. Because of these problems, tetracycline was used as an another reference compound. This compound showed strong inhibitory activities against G(+) and G(-) bacteria including K. pneumoniae and E. coli.
Comment 4:
Table 2. the names of the microorganisms must be written in italics.
Answer)
We appreciate Reviewer’s kind comments. The taxonomic designation of the microbial strains in Table 2 in the submitted manuscript was written in italics, but there seems to be a technical error in the submission process (zip files). I have corrected Table 2 again in the revised version, but I am worried that the same error will be repeated.
Comment 5:
Why ciprofloxacin is a comparator antibiotic? Explain.
Answer)
Reportedly, topoisomerase IV is the primary target of fluoroquinolones including ciprofloxacin in S. aureus, which may be due to the greater sensitivity of topoisomerase IV relative to DNA gyrase to these agents [ref. 25]. Furthermore, resistance from an altered DNA gyrase requires resistant topoisomerase IV for expression [ref. 25 and 26]. Ciprofloxacin is well known to induce resistance, and therefore, it has been frequently used as a comparator antibiotic when measuring the possibility of inducing resistance of an antibacterial active substance. In the present study, as shown in Table 3, ciprofloxacin showed potent antibacterial activities against MRSA strains S. aureus ATCC43300 and ATCC700787. However, based on multi-step (21-passage) resistance selection studies in the presence of sub-inhibitory concentrations of ciprofloxacin (a comparator antibiotic), a steady increase in MIC for ciprofloxacin against S. aureus ATCC43300 during the passage experiment was observed. These results were described in the “Discussion (page 7, lines 197-205)”.
Comment 6:
Why 4.1. General Experimental Procedures section?
Answer)
We appreciate Reviewer’s kind comments. As the reviewer pointed out, the use of “Procedures” was not considered appropriate. Thus, “4.1. General Experimental Procedures” was revised to “4.1. General Experimental Equipments”.
Comment 7:
4.3. Fermentation? Antibiotics are secondary metabolites. I do not think this is the best reference for the process of its production.
Answer)
As the reviewer pointed out, the use of this term was not considered appropriate. Thus, we revised it to “4.3. Cultivation”.
Comment 8:
How they get the 20 L of MBTI36 culture filtrate? Explain.
Answer)
We revised and provided a more detailed description in the 4.4. Extraction and Isolation section (page 8, lines 260-263): “Strain MBTI36 was cultured in GTYB broth at 28°C without shaking. After incubation for 14 days, cultures were collected from 160 flasks (each containing 125 mL of culture broth, total 20 L) and filtered through filter paper. Culture filtrate was then lyophilized and extracted three times with 5 L of MeOH.”.
Comment 9:
Why are these antibiotics (ampicillin, tetracycline, daptomycin, vancomycin, platensimycin, linezolid and ciprofloxacin) used as reference compounds and what are the highest concentrations used? (Sigma-Aldrich, St. Louis, MO, USA) were used.
Answer)
The antibiotics we used are important and frequently used clinically and in the laboratory. Ampicillin and tetracycline are mainly used as reference compounds for general antibacterial activity assays in the laboratory. In addition, daptomycin, vancomycin, platensimycin, linezolid and ciprofloxacin, which are clinically representative class antibiotics, have been used as control compounds in the evaluation of MRSA inhibitory activity assays. In the present study, the highest concentration of ampicillin and tetracycline in the antibacterial activity assays was 128 μg/mL (Table 2). In the evaluation of MRSA inhibitory activity, the highest concentration of daptomycin, vancomycin, platensimycin, linezolid, and ciprofloxacin was 32 μg/mL (Table 3).
Comment 10:
Do they use spores or cells? Authors indicate that spores were harvested from PDA plates but test compound was mixed with 10 μL of broth containing approximately 5 × 105 cfu/mL of test fungus. Explain.
Answer)
We are very sorry for the confusion over the use of the wrong technical terminology. What the reviewer pointed out was our mistake and we apologize. We used spores to antifungal activity assays. Thus, “cfu” was revised to “spores” in the 4.6. Antifungal Activity Assays section (page 9, lines 301-302): “…containing approximately 5 × 105 spores/mL of test fungus (final concentration: 5 × 104 spores /mL)…”.
Reviewer 2 Report
The manuscript entitled “Antimicrobial Activity of Chromomycins from Marine-Derived Streptomyces microflavus” investigated antibacterial activities from a marine-derived actinomycete. Genetic technique was used to characterize isolate whereas secondary metabolites were isolated and characterized using multi-pronged approach, including the NMR and mass spectrometry. Finally, antibacterial activities against Gram-positive bacteria were determined and minimum inhibitory concentration (MIC) values for all isolated/characterized compounds were determined. Authors proposed, using the passage experiment, that the MIC values don’t change considerably even after 21 passages. Overall, manuscript is well written and can be accepted after minor revision, such as fully defining the abbreviations (e.g., HRMS).
Author Response
Answer)
Thank you very much for your careful and valuable review of our manuscript. Your comments are encouraged us in doing our hardest scientific work in this research field. We made all revisions and corrections including fully defining the abbreviations, such as high-resolution fast atom bombardment mass spectrometry (HR-FAB-MS) and nuclear magnetic resonance (NMR). I hope this is the right answer for your request.
Reviewer 3 Report
I have reviewed the Manuscript titled: “Antimicrobial Activity of Chromomycins from Marine-Derived Streptomyces microflavus” and have the following comments:
The manuscript is written well and covers all the main points for a manuscript of this type. The actinobacterium has been identified by 16S rRNA gene sequencing but it has to be compared to a type strain (also called a culture with a valid name.) and not necessarily the culture which shows the closest BLAST result. The same applies to the phylogenetic trees in which the other cultures are type cultures. It is normal to make the trees using two separate program because each one uses a different algorithm so the relationship could change according to which one is used. Furthermore the trees improve their resolution if they contain an outgroup usually a species from a related genus.
The fermentation and purification are explained well enough and so too the MIC and the development of resistance. For the latter the concentration of t he ‘sub -MIC’ amount of antibiotic must be stated.
Author Response
Thank you very much for your careful and valuable review of our manuscript. Your comments are encouraged us in doing our hardest scientific work in this research field. What follows is our response to reviewer’s critique with the explanation of the changes implemented in the paper and a rebuttal when appropriate.
Comment 1:
The manuscript is written well and covers all the main points for a manuscript of this type. The actinobacterium has been identified by 16S rRNA gene sequencing but it has to be compared to a type strain (also called a culture with a valid name) and not necessarily the culture which shows the closest BLAST result. The same applies to the phylogenetic trees in which the other cultures are type cultures. It is normal to make the trees using two separate program because each one uses a different algorithm so the relationship could change according to which one is used. Furthermore the trees improve their resolution if they contain an outgroup usually a species from a related genus.
Answer)
According to the reviewer’s comments, we revised and added the sentence comparing MBTI36 with type strain Streptomyces microflavus NBRC13062T (GenBank accession number: NR_1123524) (page 2, line 66-67). Also, we provided another phylogenetic tree consisted of type culture strains (Figure 1). We hope this is the right answer for your request.
Comment 2:
The fermentation and purification are explained well enough and so too the MIC and the development of resistance. For the latter the concentration of the ‘sub -MIC’ amount of antibiotic must be stated.
Answer)
We appreciate Reviewer’s kind comments. According to the reviewer’s suggestion, the concentration of the ‘sub-MIC (0.5× the MIC determined from the previous passage)’ was stated in Figure 4 in the revised version.
Reviewer 4 Report
With the current manuscript, Shin and colleagues report the isolation and antimicrobial properties of four chromomycin derivatives from an extract obtained from a marine-derived strain of Streptomyces sp., including a previously unreported metabolite. The experimental design was properly conceived, scientific novelty relying on the isolation of chromomycin A9 as well as on the antibacterial effects towards Gram positive strains of pathogenic bacteria, including multidrug-resistant strains. While the new metabolite solely displays minor structural novelty, generated data is suitable for publication in Marine Drugs, as the authors deliver also robust data on the antibacterial activity, including the assessment of inhibitory effects towards a representative panel of multidrug resistant strains, including also multistep resistance development studies. As such, it is my opinion that the current manuscript might be considered for publication, subject to tidying up some minor issues.
While generally well written, authors are requested to consider very minor suggestions:
Line 18: Revise “…,were isolated from the secondary metabolites EtOAc extract.”.
Line 24: Revise “no resistance has been recorded developed during the 21 passages.”.
Line 32: Revise “especially Streptomyces spp.…”.
Lines 36-37: Consider “might confer unusual biosynthetic features on marine actinomycetes; thus, in comparison with their terrestrial counterparts, new bioactive secondary metabolites could be produced compared with terrestrial bacteria…”.
Line 77: Please define “GTYB” as the medium is not that common.
Line 79: Revise “…semi-preparative high-pressure performance liquid chromatography…”.
Table 2: Italicize the taxonomic designation of the microbial strains.
As solely antibacterial effects have been recorded, I would suggest revising the title of the manuscript to “Antibacterial Activity of Chromomycins from a Marine-Derived Streptomyces microflavus”.
Structure elucidation of chromomycin A9 has been properly delivered (actually, in a crystal-clear way), NMR data corroborating the proposed structure. However, authors are requested to revise “H2-2” to “H-2” (Lines 107 and 108) as it corresponds to a methine proton.
Finally, caption of Figure S2 should be revised (100 MHz).
Author Response
Thank you very much for your careful and valuable review of our manuscript. We made all revisions and corrections as far as we could. I hope this is the right answer for your request. What follows is our response to reviewer’s critique with the explanation of the changes implemented in the paper and a rebuttal when appropriate.
Minor suggestions:
Comment 1:
Line 18: Revise “…,were isolated from the secondary metabolites EtOAc extract.”.
Answer)
We appreciate Reviewer’s kind and valuable comments. According to the reviewer’s suggestion, line 18: “secondary metabolites” was revised to “EtOAc extract”.
Comment 2:
Line 24: Revise “no resistance has been recorded developed during the 21 passages.”.
Answer)
Line 24: “…developed…” was revised to “…recorded…”.
Comment 3:
Line 32: Revise “especially Streptomyces spp.…”.
Answer)
We appreciate Reviewer’s correction. Accordingly, line 32: “…Streptomyces sp…” was revised to “…Streptomyces spp…”.
Comment 4:
Lines 36-37: Consider “might confer unusual biosynthetic features on marine actinomycetes; thus, in comparison with their terrestrial counterparts, new bioactive secondary metabolites could be produced compared with terrestrial bacteria…”.
Answer)
According to the Reviewer’s suggestion, line 36-37: “might confer unusual features on marine actinomycetes; thus, new bioactive secondary metabolites could be produced compared with terrestrial bacteria” was revised to “might confer unusual biosynthetic features on marine actinomycetes; thus, in comparison with their terrestrial counterparts, new bioactive secondary metabolites could be produced”.
Comment 5:
Line 77: Please define “GTYB” as the medium is not that common.
Answer)
According to the Reviewer’s comment, line 77: the medium “GTYB” was revised to “GTYB (10 g of glucose, 2 g of tryptone, 1 g of yeast extract, and 1 g of beef extract in 1 L of artificial seawater)”.
Comment 6:
Line 79: Revise “…semi-preparative high-pressure performance liquid chromatography…”.
Answer)
We appreciate Reviewer’s correction. Accordingly, line 79: “…pressure…” was revised to “…performance…”.
Comment 7:
Table 2: Italicize the taxonomic designation of the microbial strains.
Answer)
We appreciate Reviewer’s kind comments. The taxonomic designation of the microbial strains in Table 2 in the submitted manuscript was written in italics, but there seems to be a technical error in the submission process (zip files). I have corrected Table 2 again in the revised version, but I am worried that the same error will be repeated.
Comment 8:
As solely antibacterial effects have been recorded, I would suggest revising the title of the manuscript to “Antibacterial Activity of Chromomycins from a Marine-Derived Streptomyces microflavus”.
Answer)
We appreciate Reviewer’s kind and valuable comments. According to the reviewer’s suggestion, the title of the manuscript was revised to “Antibacterial Activity of Chromomycins from a Marine-Derived Streptomyces microflavus”.
Comment 9:
Structure elucidation of chromomycin A9 has been properly delivered (actually, in a crystal-clear way), NMR data corroborating the proposed structure. However, authors are requested to revise “H2-2” to “H-2” (Lines 107 and 108) as it corresponds to a methine proton.
Answer)
According to the reviewer’s comments, we revised “H2-2” to “H-2” (Lines 110 and 111) as it corresponds to a methine proton.
Comment 10:
Finally, caption of Figure S2 should be revised (100 MHz).
Answer)
Caption of Figure S2 was revised (100 MHz).
Round 2
Reviewer 1 Report
Line 18. Write the full name and give the acronym. EtOAc?
Line 67. S. microflavus strain DR009
Line 80. Write the full name and give the acronym. MeOH?
Line 240. “To obtain single strain, colonies…” better “To obtain single strains, colonies…”
Line 255. Not necessary culture medium composition (Line 78).
Lines 258 and 261. Are they the same flasks? Volumes do not match.
Line 268. Write the full name and give the acronym. H2O-MeCN?
Line 305. Write the full name and give the acronym. RPMI?
Author Response
Thank you very much for your comments, concerning revision of our manuscript. I want to express my thanks to your careful review of our work. We provided a point-by-point response to the review’s comments with the explanation of the changes implemented in the paper. I hope this is the right answer for your request.
Comment 1:
Line 18. Write the full name and give the acronym. EtOAc?
Answer)
“EtOAc” was revised to “ethyl acetate” (line 18).
Comment 2:
Line 67. S. microflavus strain DR009
Answer)
“Streptomyces microflavus strain DR009” was revised to “S. microflavus strain DR009” (line 67).
Comment 3:
Line 80. Write the full name and give the acronym. MeOH?
Answer)
“MeOH” was revised to “methanol” (line 80).
Comment 4:
Line 240. “To obtain single strain, colonies…” better “To obtain single strains, colonies…”
Answer)
“To obtain single strain, colonies…” was revised to “To obtain single strains, colonies…” (line 240).
Comment 5:
Line 255. Not necessary culture medium composition (Line 78).
Answer)
Culture medium composition was deleted from the “4.3. Cultivation” section (line 255).
Comment 6:
Lines 258 and 261. Are they the same flasks? Volumes do not match.
Answer)
In cultivation, each seed culture (25 mL) was transferred to 100 mL of fresh GTYB broth. Thus, each containing 125 mL of culture broth. We used 160 flasks. To clear this confusion, we revised the manuscript as follows in the revised version:
Line 256. “Each seed culture was transferred directly to 100 mL of fresh GTYB broth (each flask containing 125 mL culture broth, 160 flasks used)….”.
Line 260. “Culture broths were collected from 160 flasks (total 20 L)……”.
Comment 7:
Line 268. Write the full name and give the acronym. H2O-MeCN?
Answer)
“H2O-MeCN…” was revised to “water-acetonitrile…” (Line 267-268).
Comment 8:
Line 305. Write the full name and give the acronym. RPMI?
Answer)
“RPMI…” was revised to “Roswell Park Memorial Institute (RPMI) 1640 …” (Line 302).